# The Oldest Representatives of Tree Crickets (Orthoptera: Gryllidae; Oecanthinae) from Northern Myanmar [note 1]

**DOI:** 10.3390/insects13070619

**Published:** 2022-07-11

**Authors:** Wei Yuan, Cheng-Jie Zheng, Yan-Na Zheng, Li-Bin Ma, Jun-Jie Gu

**Affiliations:** 1College of Agronomy, Sichuan Agricultural University, Chengdu 611130, China; yuanwei19980808@163.com (W.Y.); a15528302985@163.com (C.-J.Z.); 2College of Life Sciences, Shaanxi Normal University, Xi′an 710119, China; na@snnu.edu.cn

**Keywords:** mid-Cretaceous, Oecanthinae, taxonomy, new genus, new species

## Abstract

**Simple Summary:**

Two new genera and two new species of Oecanthinae (Gryllidae) are described that are from northern Myanmar amber. They are the oldest representatives of tree crickets, supporting the previous estimation of the origin of Oecanthinae by molecular data. These new findings improve our knowledge of the evolution of the Gryllidae.

**Abstract:**

The abundance of insects in Burmese amber illustrates a highly diverse insect community of the mid-Cretaceous, but the records of crickets are relatively rare. Here, we erect two new genera with two new species, *Birmanioecanthus haplostichus* gen. et sp. nov. and *Apiculatus cretaceus* gen. et sp. nov., based on two new specimens from northern Myanmar amber. These new species can be assigned to the subfamily Oecanthinae (Orthoptera: Gryllidae) by their prognathous head, slender body and metatibiae, and protibiae with large tympana. These new findings are the first and earliest fossil record of tree crickets and shed light on the evolution of Oecanithinae.

## 1. Introduction

Crickets (Gryllidae) are the most diverse group of Grylloidea, with 3364 species in 13 subfamilies, distributed all over the extant fauna. Currently, the oldest fossil record of Gryllidae can be traced back to the Early Cretaceous, represented by several species of Gryllospeculinae Gorochov, 1985 [1,2,3]. During the Cenozoic, 23 species assigned to four subfamilies were described and discovered in Europe, Asia, Africa, and America [4,5,6,7,8]. Although some molecular studies suggest that the divergence of Gryllidae (Grylloidea) might have happened during the Late Triassic to Cretaceous [9,10,11], only a few fossil records collected from the Mesozoic can be assigned to Gryllidae. There are two extinct families—Protogryllidae and Baissogryllidae of Grylloidea—erected based on the wings of compression fossils, dating from the Late Triassic to the Cretaceous [12,13]. To a certain extent, these Mesozoic grylloid species share a similar wing venation with Gryllidae but lack information on other structures of the body. At present, only 38 fossil species of Gryllidae have been reported, comprising 5 subfamilies.

Oecanthinae (tree crickets) is a unique subfamily of Gryllidae characterized by their prognathous head, slender body, and hindlegs [14]. It includes more than 300 species of 26 genera assigned to three tribes in extant fauna, widely distributed in all zoogeographical divisions [15,16]. Compared to other ground-dwelling cricket species, tree crickets (Oecanthinae) usually live in vegetation close to the ground or shrubs [17]. A molecular clock analysis based on mitogenome data indicates that Oecanthinae might split from Podoscirtinae at mid-Cretaceous (101 Mya) [10], but Cretaceous records are lacking for these subfamilies. Currently, only two Podoscirtinae species have been described from the Eocene and Oligocene of Europe [18,19].

The insects discovered from the northern Myanmar ambers are quite diverse. To date, four Grylloidea species, *Protomogoplistes asquamosus* Gorochov, 2010 (Mogoplistidae); *Pherodactylus micromorphus* Poinar, Su and Brown, 2020 (Gryllidae); *Birmaninemobius hirsutus,* Xu, Zhang, Jarzembowski, and Fang, 2020; and *Curvospurus huzhengkun,* He, 2022 (Trigonidiidae), have been described [6,20,21,22]. Here, we describe two new genera with two new species of Oecanthinae of Gryllidae from the mid-Cretaceous from the Hukawng Valley of Kachin Province in northern Myanmar. Currently, they are the first discoveries of tree crickets from Myanmar ambers, and the oldest representatives of Oecanthinae date back to the mid-Cretaceous.

## 2. Materials and Methods

The specimens were deposited at the Department of Plant Protection of Sichuan Agricultural University, Chengdu, China (SICAU). The ambers containing the tree crickets are yellow and transparent specimens from the Hukawng valley at 99 Ma, and many Grylloidea samples were collected from the site [23]. A recent study using U-Pb zircon dating determined the age to be 98.79 ± 0.62 Ma or at the Albian/Cenomanian boundary [24]. The amber containing the specimen was ground and polished on the right size.

Photographs were taken with an SZX16 microscope system and cellSens Dimension 3.2 software (Olympus, Tokyo, Japan). In most instances, incident and transmitted light were used simultaneously. All images are digitally stacked photomicrographic composites of approximately 20 individual focal planes obtained using Helicon Focus 6 (http://www.heliconsoft.com accessed on 12 May 2022) for a better illustration of the 3D structures.

The terminology of the fore wing venation follows Gorochov and Tan [25]. Sc, subcosta; R, radius; M, media; MP, posterior media, CuA, anterior cubitus; CuP, posterior cubitus; A, anals; d, dividing vein; di, diagonal vein; ch, chord vein; and o, oblique vein.

## 3. Results

Systematic paleontology.

Order Orthoptera Olivier, 1789.

Suborder Ensifera Chopard, 1920.

Supperfamily Grylloidea Laicharting, 1781.

Family Gryllidae Laicharting, 1781.

Subfamily Oecanthinae Blanchard, 1845.

Genus *Birmanioecanthus* gen. nov. Yuan, Ma et Gu

Type species. *Birmanioecanthus haplostichus* sp. nov.

Diagnosis. The head was prognathous with a broad vertex, and the compound eyes were developed and kidney-shaped; the lateral lobes of the pronotum had an obvious dark band; the subgenital plate had a pointed distal tip; the cerci were rather long with small setae; and the metatibiae only had inner dorsal spurs (autapomorphy).

Etymology. The specimen was preserved in Burmese amber; the generic name is taken from the Latin ‘Birmania’ and the generic name Oecanthus.

*Birmanioecanthus haplostichus* sp. nov. Yuan, Ma et Gu

Material. Holotype, SICAU(A)-107. Male, Burmese amber (Myanmar, Kachin Province, Hukawng Valley); mid-Cretaceous, latest Albian to earliest Cenomanian; a nearly complete adult insect with antennae, protibiae, mesotibiae, and metatibiae partly broken, and tarsus incomplete (truncated by the surface of amber); the fore wings were well preserved, and bifurcated genitalia were faintly visible at the end of the abdomen.

Diagnosis. The compound eyes were developed and were shaped like a kidney. The lateral lobes of the pronotum had an obvious dark band; the anterior and posterior margin were straight, and the posterior margin was slightly longer than the anterior. Both the fore wings and hind wings were longer than the abdomen. There were four oblique veins; mirror long and shield-like, about twice as long as the pronotum, and almost equal in length to the anterior margin of the pronotum; and the anterior margins of the mirror were approximately 60° to the distal diagonal vein. The apical field was about 1.5 times longer than the pronotum, armed with numerous cells. The metatibiae only had inner dorsal spurs. The cerci was rather long with small setae.

Etymology. The specific epithet is from the Latin ‘haplostichus’, which means a single row of spurs, used to describe the metatibia with inner dorsal spurs only.

Description. Holotype, SICAU(A)-107. Male, body form slender (Figure 1A,B).

The head was prognathous and slightly triangular; the vertex was broad, without marks or setae obviously; the frons were flat, wider than the antennae scape; the compound eyes were developed and kidney-shaped, situated near the dorsal surface of the head, protruding from the head obviously, and there was no ocellus; the antennae scape were cylindrical, located between the compound eyes, antennae filiform and gradually increased from the base to the terminal; the mouthparts and the maxillary palpus were partially obscured by emulsion; and the labial palpus was invisible (Figure 2E,F).

The pronotum transversal, anterior, and posterior margin were aequilate and slightly wider than the head; the disc had a longitudinal groove in the middle and two pairs of marks in different sizes; and the lateral lobes had a prominent dark band.

The prothoracic legs were comparatively short and thin, and the profemur and protibia had a fine setae; the procoxa was cylindrical, and the protibia had oval tympana on both sides, and one apical spur was visible; the probasitarsus was slender and shorter than the protibial; the basitarsus was without setae and apical spurs; the second tarsomere was very short; and the third tarsomere was nearly equal to the basitarsus in length. The mesothoracic legs were similar to the prothoracic legs, with mesotibia with two small apical spurs of equal length. The metathoracic legs were very long and thin, without setae on the surface; the metatibia were slightly shorter than the metafemur, covering a row of spurs on the inner dorsal margin, without outer dorsal spurs; the metatibia had five visible apical spurs; and the metabasitarsus was truncated by the surface of the amber (Figure 2A,B).

Both the fore wings and hind wings were present and developed; the tegmina was elongated, and the basal field was longer than the pronotum; the lateral field had six visible veins, which were the branches of the Sc vein; the media vein and the CuA vein intersected at the outer edge of the mirror and fused to form the lower margin of the mirror together. There were four oblique veins; the proximal vein was the shortest, and the apical vein longest. All of the oblique veins were located between the CuA vein and the stridulatory vein (S), and their two ends connected these two veins, respectively. The diagonal vein was short and almost straight, and it linked with the stridulatory vein (S) and the anterior corner of the mirror. There were three chord veins, the inner one (the CuP or the extension of the stridulatory vein) was the longest and most arc-like, the middle one (the extension of the 1A vein) was bisinuate, and the inner one (the extension of 2A vein) was the shortest and was almost straight, and there was no transverse vein linking the chord vein with mirror. The mirror was large and elongated, armed with an anterior corner that was acute and a posterior corner that was somewhat rounded. The dividing vein was slightly curved and the dividing mirror had two portions, with the posterior part slightly larger than the anterior; the apical field was long and almost as long as the mirror, with a band-like cell along the posterior margin of mirror, and several irregular cells behind it, parts of the apical field lost from the preserve. The hind wings were long and extended to the abdominal tip, were covered by the fore wings and largely invisible, and were distal with four transversal brown stripes (Figure 2C,D).

The abdomen was largely obscured by emulsion and a subgenital plate with a pointed distal tip; the cerci were incomplete; the setose was fine; and the genitalia were forked and faintly visible at the terminal.

The measurements were as follows: 9.7 mm long (measured from the head to the abdominal apex); the head was 2.56 mm long; the antennae were 1.64 mm (left)/7.81 mm (right) preserved; the fore wing length of 7.96 mm was preserved; the hind wing length was 10.99 mm; the profemur was 2.46 mm; the protibia was 2.33 mm; the mesofemur was 2.12 mm; the mesotibia was 2.11 mm; the metafemur was 5.85 mm; and the metatibia was 5.79 mm and preserved (left).

Geuns Apiculatus gen. nov. Yuan, Ma et Gu

Type species. *Apiculatus cretaceus* sp.nov.

Etymology: The genus name derives from the Latin ‘Apiculatus’ used to describe its needle-like ovipositor.

Diagnosis. The head was prognathous with a narrow vertex and frons, the pronotum was a light color, the metatibia had two rows of tiny spines and three inner dorsal spurs (autapomorphy), and the ovipositor was needle-like and without a specialized tip.

*Apiculatus cretaceus* sp. nov. Yuan, Ma et Gu

Material. Holotype, SICAU(A)-113. Female, Burmese amber (Myanmar, Kachin Province, Hukawng Valley); mid-Cretaceous, latest Albian to earliest Cenomanian; a nearly complete adult insect with antennae, mesothoracic and metathoracic legs that were partly broken (truncated by the surface of amber), and fore wings that were well preserved but mostly hidden by cracks; the ovipositor was complete (Figure 3A,B).

Etymology: After the Cretaceous Period.

Diagnosis. The protibia had small oval tympana on both sides and was apically armed with one visible spur; the probasitarsus was slender and slightly shorter than the protibia. The metathoracic legs very long and narrow, without setae on the surface; the metatibia was incomplete and was armed with tiny spines, and the three inner dorsal spurs were visible. The cerci was rather long and distinctly longer than the ovipositor, and the setose was fine. The ovipositor was needle-like with a pointed terminal.

Description. Holotype, SICAU(A)-113. Female, body form slender (Figure 3A,B).

The head was prognathous and slightly triangular; it had a vertex narrow, without marks or setae; the frons were exceedingly narrow, conspicuously narrower than the antennae scape; compound eyes developed, small and oval-shaped, obviously protruding from the head; there was no ocellus; the antennae scape was cylindrical and black, located between the compound eyes and the antennae filiform and partly truncated by the surface of amber; the maxillary palpus was slender, the end joint was slightly broad, and the labial palpi was relatively short (Figure 4A,B).

The pronotum was slightly trapezoidal, the anterior and posterior margins were aequilate and slightly wider than head; and the disc was mostly hidden by cracks and a lateral lobe with a light color.

The prothoracic legs were short and thin, and the profemur and protibia had fine hairs. The protibia had long and oval tympana on both sides, and the apex had one visible spur. The probasitarsus was slender and shorter than the protibia; the basitarsus was without the setae and apical spurs; the second tarsomere was very short; and the third tarsomere was significantly shorter than the basitarsus, about half of the basitarsus. The mesothoracic legs were truncated by the surface of amber, and most parts were invisible. The metathoracic legs were very long and thin, without setae on the surface; the metatibiae were incomplete, with two rows of tiny spines and three inner dorsal spurs visible; and the metabasitarsus was truncated by the surface of the amber (Figure 4C–F).

Both the fore wings and hind wings were present and developed and were obviously longer than the abdomen; the fore wings had several parallel veins visible, most hidden by cracks.

The abdomin was stubby, the subgenital plate had a pointed distal tip, the cerci were rather long, the setose was fine, and the ovipositor was needle-like with a pointed terminal (Figure 3C,D).

The measurements were as follows: 11.6 mm long (measured from head to abdominal apex); the head was 2.52 mm; the antennae were 9.69 mm (left)/6.68 mm (right) preserved; the fore wing length was 15.46 mm; the ovipositor was 7.31 mm; the profemur was 3.46 mm, the protibiae was 2.69 mm; the mesofemur was 2.18 mm preserved; the metafemur was 8.20 mm; and the metatibia was 6.82 mm preserved (right).

## 4. Discussion

These new findings can be attributed to the subfamily Oecanthinae based on the following characteristics: head prognathous; mouth parts directed forwards; protibiae with large tympana; and body and hindlegs slender. *Birmanioecanthus haplostichus* sp. nov. shares with the tribe Paroecanthini Gorochov, 1986, a dorsoventrally depressed head with a flattened dorsal area, but it can be excluded from the Paroecanthini by its very large eyes, its metatibiae without spines, and the shape of its mirror. Four oblique veins are present in *B**. haplostichus* sp. nov. This characteristic contrasts with the tribes Oecanthini Blanchard, 1845 and Xabeini Vickery and Kevan, 1983, whose oblique veins are always less than three but who is quite similar to the tribe Paroecanthini Gorochov, 1986, which also has more than four oblique veins. A further similarity of fore wings between the new taxon and the tribe Paroecanthini is the length, size, and number of wing cells found in the apical field of the fore wings. On the other hand, the apical field of the fore wings of the tribes of Oecanthini and Xabeini are extremely brief, either totally devoid of wing cells or only armed with a very small number. Additionally, the mirror of the new species *B. haplostichus* resembles those of the genera of *Ectotrypa* Saussure, 1874; *Perutrella* Gorochov, 2011; and *Stenoecanthus* Chopard, 1912, and its protibial tympanum resembles that of the genus *Paroecanthus* Saussure, 1859. However, the new fossil differs from these genera in that it only has inner dorsal spurs on its metatibia. As a result, we create the new genus *Birmanioecanthus* for the subfamily Oecanthinae and designate *B. haplostichus* sp. nov. as the type species.

*Apiculatus cretaceus* sp. nov. shares certain characteristics with the tribes of Oecanthini and Xabeini, and the genus of *Stenoecanthus* Chopard, 1912. The new species has a narrow hind femur and a lengthy fore tarsus that is just shorter than the protibia. These characteristics are shared by a species belonging to the tribes Oecanthini and Xabeini. The cerci of the new fossil resemble the species of *Stenoecanthus* Chopard, 1912. However, the ovipositor and metatibiae of the *Apiculatus* set it apart. Oecanthini, Xabeini, and *Stenoecanthus* have apically obtuse ovipositors, while the new species has a sharp ovipositor. Other species of Oecanthinae have metatibial dorsal spurs on both sides or do not have any at all; however, the new taxa only have a row of inner spurs. *Apiculatus cretaceus* sp. nov. shares with Paroecanthini a somewhat dorsoventrally depressed head and not very large eyes located near the dorsal surface of the head, but it can be distinguished from the tribe by a needle-like ovipositor and metatibiae with only inner dorsal spurs. The size and shape of the body and metatibiae of *A. cretaceus* sp. nov. are similar to those of *B. haplostichus* sp. nov., but the pronotum lateral lobe lacks a dark band; the metatibiae have two rows of tiny spines and fewer inner dorsal spurs, making them different from those of *B. haplostichus* sp. nov. Although *A. cretaceus* sp. nov. is only recorded by one female specimen, the differences mentioned above do not support the sexual dimorphism of the species. *Pherodactylus micromorphus* (Poinar, Su and Brown, 2020) is the first gryllid insect described from Myanmar amber; its modified protibiae can be easily distinguished from the two new species described here. The two new species described here have only inner dorsal spurs. Therefore, based on a combination of these characters, the erection of these two new genera and species is assigned to Oecanthinae; *Birmanioecanthus haplostichus* sp.nov. and *Apiculatus cretaceus* sp.nov are justified and could be the earliest diverging lineages in Oecanthinae.

Modern Gryllidae are extremely diverse with various habitats, including ground-dwelling and arboreal types. While the current studies indicate that infraorders Tettigoniidea and Gryllidea might split earlier than in the Permian [9,10,11] or during the Late Triassic [26], the fossil evidence is scarce. The earliest and definite Gryllidae were found in the Early Cretaceous Crato Formation of Brazil, the Dzun–Bain Formation of Mongolia, and the Weald Clay Formation of England [1,2,3,27]; they were described as several species of an extinct subfamily Gryllospeculinae based on the fore wings from compression fossils. Although a molecular study has speculated that the subfamilies within Gryllidae may have diverged during the Middle Jurassic (Aalenian to Bajocian) [11], only 4 of 13 subfamilies of Gryllidae have a fossil record, which is additionally limited from the Eocene to Miocene. *Pherodactylus micromorphus* is a lately described fossil species attributed to Gryllidae from the Kachin amber, but its paddle-like protibiae, well-developed protibial spurs, and significantly elongated pronotum distinguish it from other crickets; meanwhile, its body shape and digging legs are similar to *Burmagryllotalpa longa* (Wang, Lei, Zhang, Xu, Fang, and Zhang, 2019) and *Tresdigitus rectanguli* (Xu, Fang, and Wang, 2020), which were both assigned to Gryllotalpidae. The systematic placement of *Pherodactylus micromorphus* needs to be further considered. Thus, the two new species described here are the most well-preserved and definite fossil Gryllidae from the Cretaceous, and we have the opportunity to clarify their taxonomic position. These new findings indicate that the oldest representatives of the subfamily Oecanthinae might appear no later than 100 Mya, which is largely consistent with the divergence time estimation based on the molecular data [10]. Meanwhile, it suggests that we are likely to find closely related groups, such as Podoscirtinae and Hapithinae [10,28], during the mid-Cretaceous. Although it is hard to observe the structures associated with climbing on the plants from these ambers, their slender thorax legs suggest that they were most likely adapted to live in vegetation close to the ground rather than to be ground-dwellers. It has already been found that Elcanidae, Trigonidiidae, Gryllidae, Mogoplistidae, Gryllotalpidae, and Tridactyllidae, in the northern Myanmar ambers [19,20,21,22,29,30], reflect a broad adaption of the ecological niche of orthopterans in the Myanmar amber biota.

## 5. Conclusions

Based on the morphological characters discussed above, two new genera and two new species, *Birmanioecanthus haplostichus* gen. et sp. nov. and *Apiculatus cretaceus* gen. et sp. nov., are reported here. They can be assigned to the subfamily Oecanthinae of Gryllidae. As it stands, these new species are the oldest representatives of Oecanthinae, implying that this group might originate no later than the mid-Cretaceousc (ca. 100 Mya). So far, the Burmese amber biota has reported 22 species of Orthoptera from Elcanidae, Gryllidae, Trigonidiidae, and Gryllotalpidae, demonstrating high species diversity.

## Figures and Tables

**Figure 1 insects-13-00619-f001:**
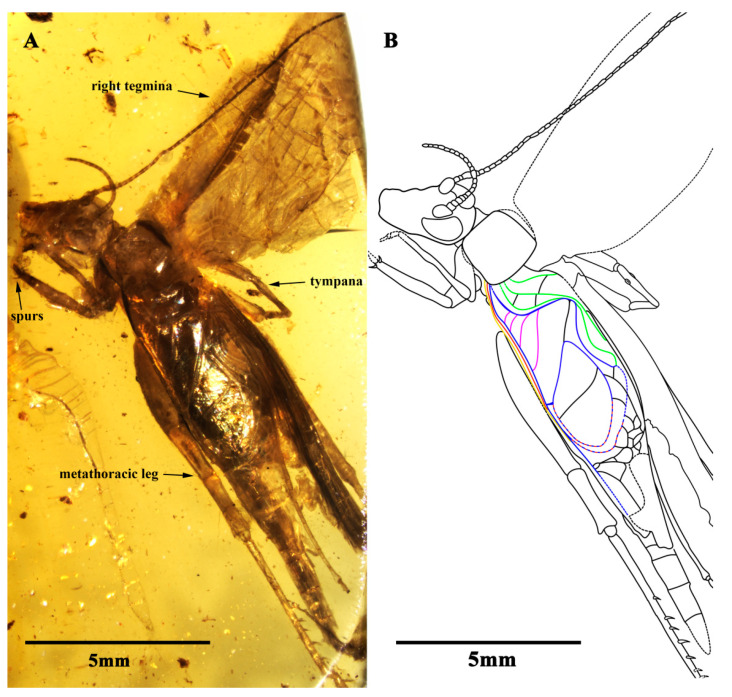
Habitus of *B. haplostichus* sp. nov.: (**A**) photograph of dorsal view, (**B**) line drawing of dorsal view. (Description of the color lines in (**B**): the yellow line represents radius vein (R); the red line represents media vein (M); the blue lines represent cubitus veins (CuA, CuP); the green lines represent anals veins (1A, 2A and 3A); and the pink lines represent oblique vein (o)).

**Figure 2 insects-13-00619-f002:**
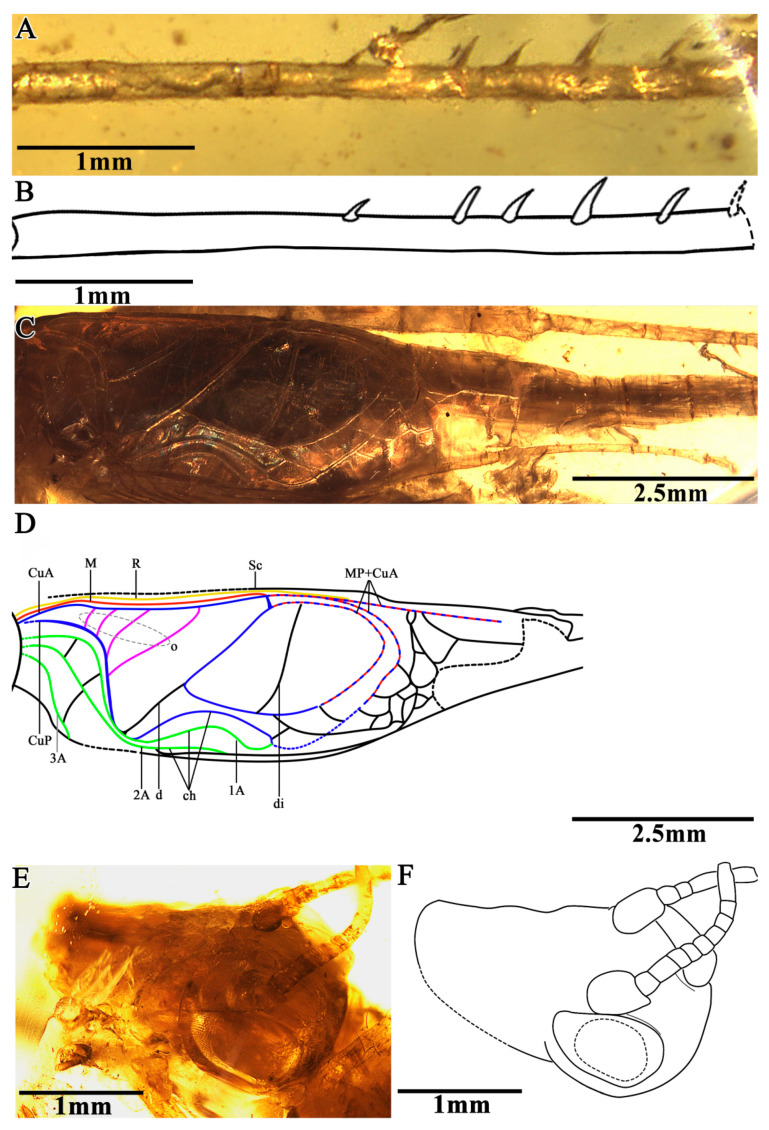
Details of *B. haplostichus* sp. nov.: (**A**) photograph of left metatibia, (**B**) line drawing of left metatibia, (**C**) photograph of left tegmina, (**D**) line drawing of left tegmina, (**E**) photograph of head, and (**F**) line drawing of head. (Description of the color lines in (**D**): the yellow line represents radius vein (R); the red line represents media vein (M); the blue lines represent cubitus veins (CuA, CuP); and the green lines represent anals veins (1A, 2A and 3A); the pink lines represent oblique vein (o)).

**Figure 3 insects-13-00619-f003:**
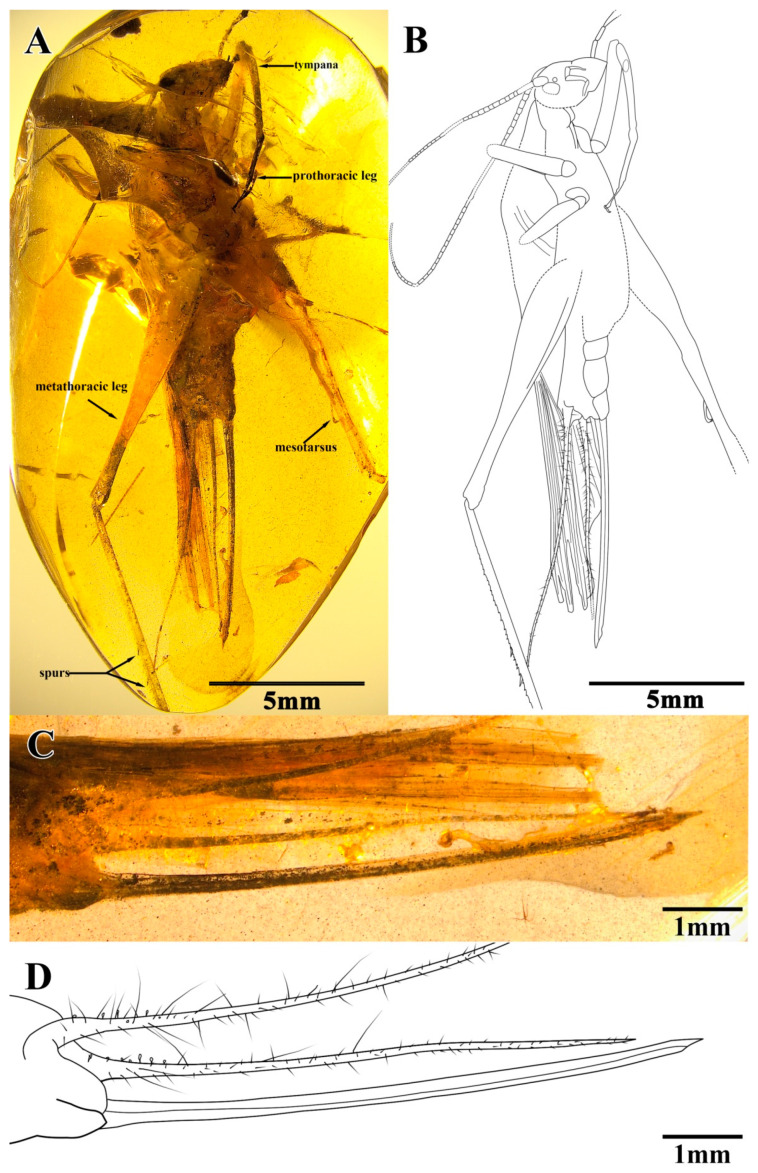
*A. cretaceus* sp. nov.: (**A**) photograph of lateral view, (**B**) line drawing of lateral view, (**C**) photograph of ovipositor, and (**D**) line drawing of ovipositor.

**Figure 4 insects-13-00619-f004:**
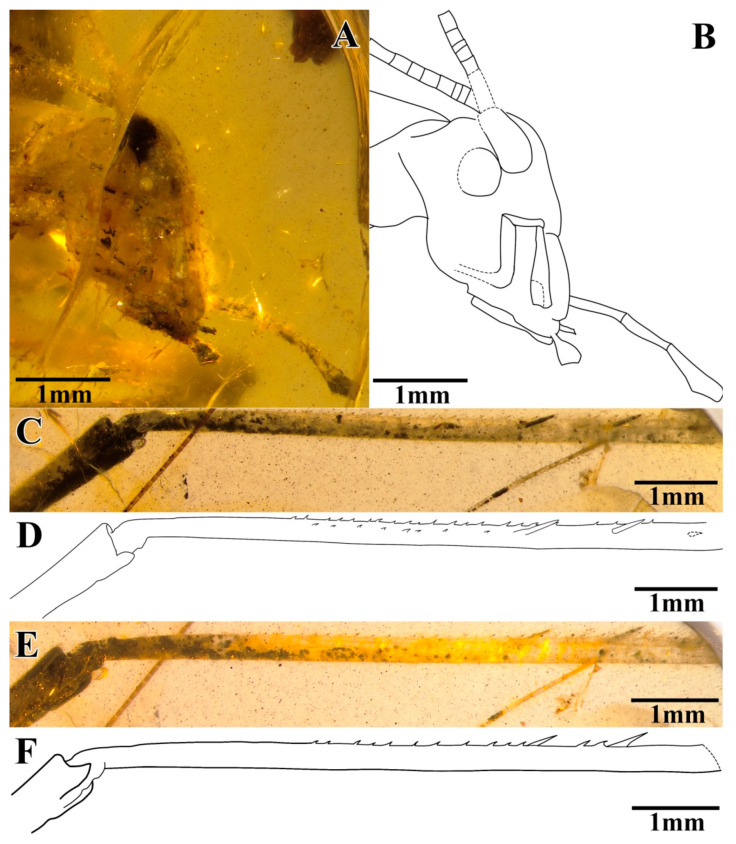
Details of *A. cretaceus* sp. nov.: (**A**) photograph of head, (**B**) line drawing of head, (**C**) photograph of metatibia (inner side), (**D**) line drawing of metatibia (inner side), (**E**) photograph of metatibia (outer side), and (**F**) line drawing of metatibia (outer side).

## Data Availability

The data presented in this study are available on request from the corresponding author.

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
