# Peer review of "The Oldest Representatives of Tree Crickets (Orthoptera: Gryllidae; Oecanthinae) from Northern Myanmar†"

_insects, 2022, doi:10.3390/insects13070619_

Round 1
Reviewer 1 Report
lateral lobes of pronotum with the prominent dark band
lateral lobes of pronotum with a prominent dark band
please, could you precise the autapomorphies of the new genera in the diagnoses ?
situated near dorsal surface of the head
situated near dorsal surface of head
ocellus absent
no ocellus
located between the compound eyes
forewings preserved well but most hidden
forewings well preserved but most hidden
in the drawing of wing, you should be able to see the vein M+CuA and the distal parts of M, CuA+CuPA(alpha)
by the way, which venation terminology do you use?
body slender and metatibia with spurs
all the crickets have spurs on metatibia, no?
which characters are synapomorphies of the subfamily, please, precise
shares with tribe Paroecanthini head
shares with tribe Paroecanthini a head
stridulatory apparatus of male tegmina with rather numerous oblique 163 veins
please, precise what means 'rather numerous', compared to what (and a reference would be nice to add here)
Figure 4. A. photograph of head
it is of which species?
are justified and could be the earliest diverging lineages
are justified and could be the earliest diverging lineages
what about outline, precise
the forewings of B. haplostichus sp. nov. have long apical field and metatibia only with inner dorsal spurs
B. haplostichus sp. nov. has forewings with a long apical field and metatibia only with inner dorsal spurs
are justified and could be the earliest diverging lineages
are justified. They could be the earliest diverging lineages
Gryllidae is extremely diverse with various habitats in the modern biota
Modern Gryllidae are extremely diverse with various habitats
Author Response
Dear Colleague:
Thank you very much for your quick, kind and valuable comments and suggestions. We have modified them as following:
Reviewer 1:
“lateral lobes of pronotum with the prominent dark band”
“lateral lobes of pronotum with a prominent dark band”
please, could you precise the autapomorphies of the new genera in the diagnoses?
Answer: We have emended the diagnosis and indicated the autapomorphies.
“situated near dorsal surface of the head”
“situated near dorsal surface of head”
Answer: thanks, we deleted ‘the’.
“ocellus absent”
“no ocellus”
Answer: thanks, we changed.
“forewings preserved well but most hidden”
“forewings well preserved but most hidden”
Answer: thanks, we changed.
in the drawing of wing, you should be able to see the vein M+CuA and the distal parts of M, CuA+CuPA(alpha), by the way, which venation terminology do you use?
Answer: we added a description of the wing venation and the terminology in methods section.
“body slender and metatibia with spurs”, all the crickets have spurs on metatibia, no? which characters are synapomorphies of the subfamily, please, precise.
Answer: we deleted the character of spurs. To date, there are no phylogenetic studies based on morphology containing all subfamilies of Gryllidae, and clarifying the synapomorphies of Oecanthinae. Although we follow the composition of Oecanthinae by Denadai de Campos L. & Desutter-Grandcolas (2020) and Cigliano et al. (2022), this attribution is based on the molecular analysis of Chintauan-Marquier et al. (2015), the study only included on one of two tribes of Oecanthinae, the composition of subfamily Oecanthinae is controversial currently. Generally, subfamily Oecanthinae (including tribes Oecanthini, Xabeini and part of Paroecanthini) can be identified by the followings: head prognathous, mouth parts directed forwards; protibia with large tympana; body slender.
“shares with tribe Paroecanthini head”
“shares with tribe Paroecanthini a head”
Answer: thanks, we added ‘a’.
“stridulatory apparatus of male tegmina with rather numerous oblique veins”, please, precise what means 'rather numerous', compared to what (and a reference would be nice to add here)
Answer: thanks, we we have modified.
“Figure 4. A. photograph of head”, it is of which species?
Answer: thanks, we modified.
what about outline, precise
Answer: thanks, it is redundant word, we have deleted it.
“the forewings of B. haplostichus sp. nov. have long apical field and metatibia only with inner dorsal spurs”
“B. haplostichus sp. nov. has forewings with a long apical field and metatibia only with inner dorsal spurs”
Answer: thanks, we changed
“are justified and could be the earliest diverging lineages”
“are justified. They could be the earliest diverging lineages”
Answer: thanks, we changed.
“Gryllidae is extremely diverse with various habitats in the modern biota”
“Modern Gryllidae are extremely diverse with various habitats”
Answer: thanks, we changed.
Reviewer 2 Report
Fossil records of true crickets from the Mesozoic are rare. Yuan and colleagues describe two new genera with two new species, assigned to the subfamily Oecanthinae (Orthoptera: Gryllidae), from north Myanmar amber fill the gap in the fossil record. The fossils are well described (although detailed to be improved), and the systematic placements are supported by the morphological data given. I have a few comments about the line drawings and format.
Line drawing problem:
Fig. 1B, the head, protibial (left) and wing venation morphologies need more precision;
Fig. 2A,B, the spines in the line drawing (B) seem to be fewer than they should be (A) in the photo; please check;
Line 32: might happened during the Late Triassic to Middle Jurassic [9,10],
Here please also cite the recent work on the orthopteran timetree by Song et al. (2020) [ref. 19 in the Ms], as this study suggested a much later origin of Gryllidae.
Lines 71-72: change ‘embraced’ to ‘preserved’; Oecanthus in italics.
Line 81: Specific Description section: detailed measurements [head, antenna, etc.] and colouration should be given here. Same for the other species.
Line 121: ‘cretaceus’ is not Latin, please reword it to ‘crētāceus’;
Line 159: what are the synapomorphies for the subfamily Oecanthinae? Please cite a morphological phylogenetic work and specify which are the synapomorphies that support the systematic placement of the fossils.
Lines 187-188: Tettigoniidea and Gryllidea?? Are you supposed to mean the families or superfamilies. The current spellings are wrong.
Lines 211-212: need rewording.
Author Response
Dear Colleague:
Thank you very much for your quick, kind and valuable comments and suggestions. We have most of your suggestions, please see the following:
Reviewer 2:
Line drawing problem:
Fig. 1B, the head, protibial (left) and wing venation morphologies need more precision;
Fig. 2A,B, the spines in the line drawing (B) seem to be fewer than they should be (A) in the photo; please check;
Answer: thanks, we have revised the figures.
Line 32: might happened during the Late Triassic to Middle Jurassic [9,10],
Here please also cite the recent work on the orthopteran timetree by Song et al. (2020) [ref. 19 in the Ms], as this study suggested a much later origin of Gryllidae.
Answer: thanks, We modified this sentence.
Lines 71-72: change ‘embraced’ to ‘preserved’; Oecanthus in italics.
Answer: thanks, we changed.
Line 81: Specific Description section: detailed measurements [head, antenna, etc.] and colouration should be given here. Same for the other species.
Answer: thanks, we have added the measurements.
Line 121: ‘cretaceus’ is not Latin, please reword it to ‘crētāceus’;
Answer: thanks, we corrected it.
Line 159: what are the synapomorphies for the subfamily Oecanthinae? Please cite a morphological phylogenetic work and specify which are the synapomorphies that support the systematic placement of the fossils.
Answer: To date, there are no phylogenetic studies based on morphology containing all subfamilies of Gryllidae, and clarifying the synapomorphies of Oecanthinae. Although we follow the composition of Oecanthinae by Denadai de Campos L. & Desutter-Grandcolas (2020) and Cigliano et al. (2022), this attribution is based on the molecular analysis of Chintauan-Marquier et al. (2015), the study only included on one of two tribes of Oecanthinae, the composition of subfamily Oecanthinae is controversial currently. Generally, subfamily Oecanthinae (including tribes Oecanthini, Xabeini and part of Paroecanthini) can be identified by the followings: head prognathous, mouth parts directed forwards; protibia with large tympana; body slender.
Lines 187-188: Tettigoniidea and Gryllidea?? Are you supposed to mean the families or superfamilies. The current spellings are wrong.
Answer: Here we mean the infraordersTettigoniidea and Gryllidea, the spellings are correct.
Lines 211-212: need rewording.
Answer: thanks, we have reworded this sentence.
